# Diagnostic and prognostic significance of premature ventricular complexes in community and hospital-based participants: A scoping review

Sukardi Suba [1]*, Kirsten E. Fleischmann[2], Hildy Schell-Chaple[3], Priya Prasad[4], Gregory M. Marcus[2], Xiao Hu[5], Michele M. Pelter [6]*

1 School of Nursing, University of Rochester Medical Center, Rochester, New York, United States of America, 2 Department of Medicine, Division of Cardiology, School of Medicine, University of California, San Francisco, San Francisco, California, United States of America, 3 Institute for Nursing Excellence, University of California, San Francisco, Medical Center, San Francisco, California, United States of America, 4 Department of Medicine, Division of Hospital Medicine, School of Medicine, University of California, San Francisco, San Francisco, California, United States of America, 5 School of Nursing, Duke University, Durham, North Carolina, United States of America, 6 Department of Physiological Nursing, University of California, San Francisco, California, United States of America

* sukardi_suba@urmc.rochester.edu (SS); michele.pelter@ucsf.edu (MMP)

**Data Availability Statement:** All relevant data are within the manuscript and its Supporting Information files.

## Abstract

### Background

While there are published studies that have examined premature ventricular complexes (PVCs) among patients with and without cardiac disease, there has not been a comprehensive review of the literature examining the diagnostic and prognostic significance of PVCs. This could help guide both community and hospital-based research and clinical practice.

### Methods

Scoping review frameworks by Arksey and O'Malley and the Joanna Briggs Institute (JBI) were used. A systematic search of the literature using four databases (CINAHL, Embase, PubMed, and Web of Science) was conducted. The review was prepared adhering to the Preferred Reporting Items for Systematic Reviews and Meta-Analysis Extension for Scoping Review (PRISMA-ScR).

### Results

A total of 71 relevant articles were identified, 66 (93%) were observational, and five (7%) were secondary analyses from randomized clinical trials. Three studies (4%) examined the diagnostic importance of PVC origin (left/right ventricle) and QRS morphology in the diagnosis of acute myocardial ischemia (MI). The majority of the studies examined prognostic outcomes including left ventricular dysfunction, heart failure, arrhythmias, ischemic heart diseases, and mortality by PVCs frequency, burden, and QRS morphology.

**Funding:** SS received ECG Monitoring Research Pre-Doctoral Fellowship from UCSF School of Nursing. The funder had no role in study design, data collection and analysis, decision to publish, or preparation of the manuscript.

**Competing interests:** The authors have declared that no competing interests exist.

## Conclusions

Very few studies have evaluated the diagnostic significance of PVCs and all are decades old. No hospital setting only studies were identified. Community-based longitudinal studies, which make up most of the literature, show that PVCs are associated with structural and coronary heart disease, lethal arrhythmias, atrial fibrillation, stroke, all-cause and cardiac mortality. However, a causal association between PVCs and these outcomes cannot be established due to the purely observational study designs employed.

## Introduction

Premature ventricular complexes (PVCs) are early depolarizations of myocardial cells that originate in the ventricle (right or left) and are caused primarily by impulse formation disorder (enhanced automaticity or triggered activity) [1–3] or reentry mechanisms of myocardial tissues [1, 4–6]. In hospitalized patients, PVCs are one of the most common arrhythmias seen on both standard 12-lead electrocardiograms (ECGs) or during continuous ECG monitoring [7–9]. One hospital-based study found that there were over 854,901 PVC alarms during the one-month study period, representing 33% of the over 2.5 million total alarms or 18 PVCs/monitoring hour per patient [7]. Among community-based participants enrolled in large cohort studies, the occurrence rate of PVCs was found to be 1% to 4% [10–12]. PVCs are more prevalent in individuals with structural heart disease (SHD), suggesting they may be a marker of cardiac pathology in some subjects [13].

In the late 1960s, when ECG monitoring was first introduced in the hospital setting, PVCs were carefully monitored for and treated, particularly in patients with acute coronary syndromes, because PVCs were considered a potential precursor to lethal arrhythmias (i.e., ventricular tachycardia [VT] and/or ventricular fibrillation [VF]) [14]. Following two decades of treating PVCs in hospitalized patients without empirical evidence, the landmark Cardiac Arrhythmia Suppression Trial (CAST) [15] tested the hypothesis that pharmacological suppression of PVCs would reduce the incidence of arrhythmic death in post-myocardial infarction (MI) patients. Surprisingly, preliminary data from the CAST study showed that pharmacologic suppression of PVCs using encainide or flecainide (class IC antiarrhythmics) was associated with increased mortality when compared with placebo [15]. This finding from the interim analysis led to early termination of the study and a shift away from routine aggressive treatment of PVCs in clinical practice. Interestingly, even though PVCs are not typically treated in the hospital setting, bedside ECG monitors used in both the intensive care and telemetry unit settings are often configured to alarm (audible and inaudible message alerts) for PVCs. Not only is this considered potential over monitoring [16, 17], but this practice can cause alarm burden in clinicians and thus, contribute to alarm fatigue [7, 18]. The Practice Standards for ECG monitoring in hospital settings state that the benefit of continuous PVC monitoring is less well-established (class: IIb), and unfortunately, there is a paucity of literature regarding the potential relevance of PVCs among hospitalized patients (level of evidence: C) [17]. Hence, guidance on how best to monitor and/or manage PVCs in hospitalized patients is mostly unknown.

In the outpatient setting, there is debate as to whether PVCs are generally benign or serve as a marker of risk for various cardiovascular diseases such as left ventricle (LV) dysfunction, cardiomyopathies, or MI [19, 20]. Although published consensus and practice guidelines provide

guidance for the management of PVCs [21–25], the recommendations come from observational data, thus, confirmatory cause and effect regarding PVCs on clinical outcomes is still unknown.

Three meta-analyses that examined community-based participants without known cardiac disease identified PVCs as a predictor of mortality [13, 26, 27]. In one meta-analysis, there was an association between the presence of PVCs and an increased risk for all-cause and cardiovascular mortality [13]. Another study showed that frequent PVCs were an independent risk factor for sudden and overall cardiac death [26]. Moreover, in patients undergoing exercise stress testing, the presence of PVCs was correlated with a higher risk for mortality [27]. While these meta-analyses show that PVCs were associated with an increased risk for all-cause and cardiovascular mortality, they included only outpatient and community-based participants without cardiac disease. Of note, this same evidence is lacking in other important groups such as hospitalized patients, asymptomatic patients with cardiac disease, and/or patients with implantable cardioverter-defibrillator (ICD). Therefore, there is a need to establish a clearer understanding of the diagnostic and prognostic implications of PVCs in both community and hospital settings and whether there are differences based on the presence of underlying cardiac disease. This would not only help guide clinical practice (in- and out-patient), but shed light on whether PVCs are an important marker in patients at risk for the development of cardiac disease and/or adverse outcomes. The purpose of this scoping review was to evaluate published studies to date that have examined the diagnostic and prognostic significance of PVCs across different care settings (i.e., community, hospital) and among various adult patient populations (i.e., with-, without heart disease). Of note, this scoping review included historical studies as far back as the 1970's with the goal of understanding shifts in patient populations of study, clinical and prognostic importance and the analysis of newer cardiac pathologies associated with PVCs.

## Materials and methods

Our scoping review protocol has been registered in the Open Science Framework (OSF), DOI: https://doi.org/10.17605/OSF.IO/GAVT2, which includes a published full-text [28]. The review followed the scoping review framework of Arksey and O'Malley [29] and the Joanna Briggs Institute (JBI) Methodology for Scoping Reviews [30]. This manuscript was prepared using the Preferred Reporting Items for Systematic Reviews and Meta-Analysis Extension for Scoping Review (PRISMA-ScR): Checklist and Explanation [31], and can be found in S1 Table. The methods of the present scoping review followed our published protocol [28] with a few minor changes, which will be described below.

### Eligibility

We included primary studies that collected quantitative data that were published in English. We included studies that met the Population, Concept, and Context (PCC) framework set forth by the JBI scoping review methodology (Table 1) [30]. The gold standard test for diagnosing PVCs is the ECG [25]; therefore, all of the included studies used some type of ECG device/method to diagnose PVCs (e.g., standard 12-lead ECG, Holter, or bedside monitor).

### Information sources and search strategy

The preliminary search was done by SS with a pilot review and guidance on the content and review methodology from MMP. After both investigators were confident that the search and review process was well established, SS performed the literature search and selected the studies for inclusion with validation by MMP. Four electronic databases were searched: the

**Table 1. Population, Concept, and Context (PCC) eligibility criteria used to identify premature ventricular complex (PVC) studies for the scoping review [28], adapted from the Joanna Briggs Institute (JBI) [30].**

| Eligibility Component | Criteria |
|---|---|
| Participants | Adults 19 years of age or older who had premature ventricular complexes (PVCs) diagnosed. Studies that examined exercise-induced PVCs (EiPVCs) were included if the procedure was performed during hospitalization or a clinic visit as part of a screening or diagnostic test. |
| Concept | The focus of this scoping review was to assess the diagnostic (i.e., identification of a specific condition or disease) and prognostic importance (i.e., patient outcomes) of PVCs. Studies that examined the combination of both PVCs and other arrhythmias were excluded. Finally, studies that assessed electrocardiographic (ECG) features of PVCs (e.g., QRS duration, morphology) were included only if the study also examined the correlation of such features with patient outcomes. |
| Context | Studies conducted in both outpatient and inpatient settings were included. There were no exclusions based on geographical location or demographics (e.g., race, ethnicity, sex, etc.), or year of publication. |

Cumulative Index to Nursing and Allied Health Literature (CINAHL), Embase, PubMed, and the Web of Science Core Collection, and included publications without limitations on publication date. Once the electronic search had been completed, the reference lists for all of the available articles were carefully searched manually to ensure there were no studies excluded during the electronic search. The detailed database search strategy is shown in S2 Table.

## Selection process

The title for each study identified was exported into EndNote X8 (Clarivate Analytics, PA, USA) in order to identify and remove duplicates. These references were then exported to Microsoft Excel (version 2016; Microsoft, Redmond, WA) for a secondary duplicate screening process. The remaining list was carefully screened to ensure the established eligibility criteria were met. The first author (SS) conducted the screening, selection, and review process independently. The last author (MMP) provided oversight to ensure the established criteria were met for the selected studies and to ensure that the extracted data from the included studies were captured accurately. Fig 1 is the PRISMA flow diagram detailing our study selection process [32].

## Data charting

A data extraction form was developed by SS and MMP [28]. Minor modifications to the data extraction form were made following a pilot study. For example, data on PVC criteria (e.g., simple, complex, or frequent PVCs) and the ECG annotation method were originally included on the data extraction form. However, after reviewing over half of the studies, we determined that the majority of the studies did not report these data elements; hence, these variables were not extracted. The first author (SS) performed the data extraction process, with consultation from MMP whenever data were unclear.

## Data extracted and synthesis of the results

Data extracted for this scoping review included study characteristics (i.e., country of origin, year, aims, design, setting, patient population, sampling criteria/size, age, and sex), study methodology (i.e., ECG method, follow-up period, and analysis), and key findings/outcomes. While we examined studies for ethnicity and race, very few provided these data. Because the purpose of this present review was to gather evidence on the prognostic and clinical significance of PVCs, only studies that performed statistical analyses were included in the scoping

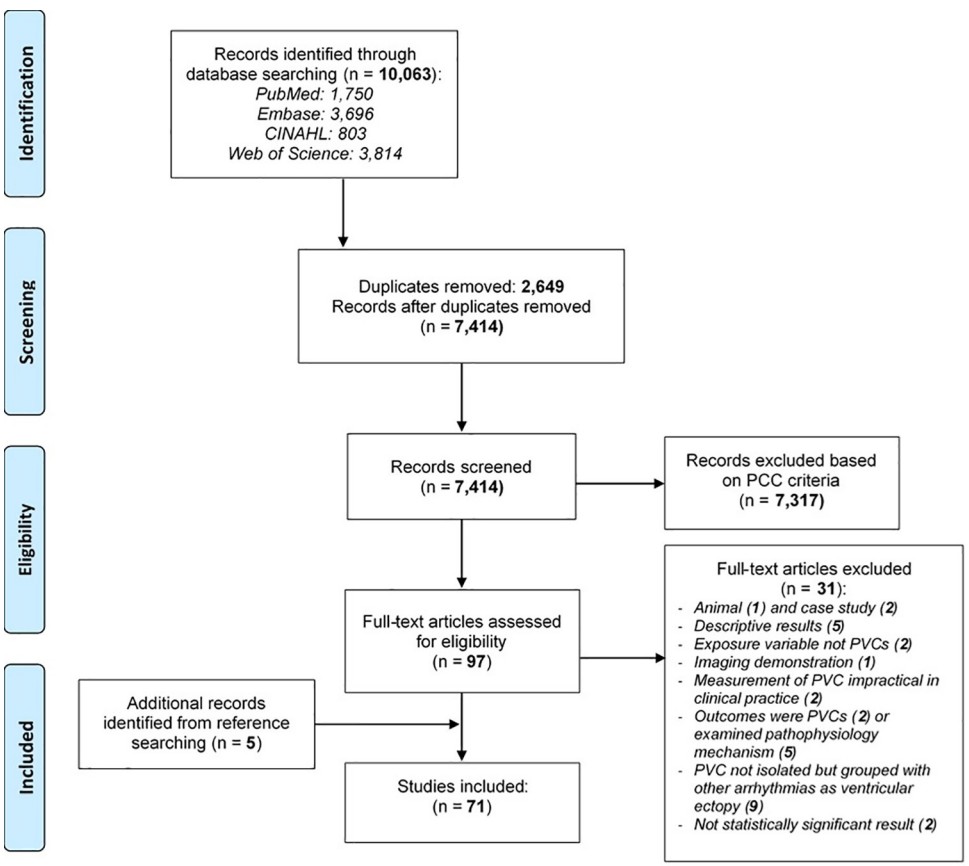

**Fig 1. Study selection process and results (adopted from PRISMA 2009 flow diagram [32]).**
CINAHL = Cumulative Index to Nursing and Allied Health Literature; PCC = Participants-Concept-Context.

review. The data from the studies included in our review are presented in tables. We grouped the evidence primarily based on the setting (outpatient versus hospital-based), presence or absence of cardiac disease(s) and patient outcomes. Key findings are synthesized and described in detail in the narrative text.

## Critical appraisal/Publication bias

We were unable to perform a critical appraisal of the individual source evidence included in this review as initially planned because there was significant methodological heterogeneity across the studies. Nevertheless, since the goal of this scoping review was to "map" the existing evidence regarding the significance of PVCs regardless of the methodological quality, a critical appraisal is not required. This is aligned with the PRISMA-ScR guidelines [31], hence, our approach is within the guidelines of a well-designed review of the literature.

## Results

The literature search was conducted in January 2020, and resulted in a total of 10,063 titles from the four databases searched. Five additional titles were identified after searching the cited references. After a careful screening process using the outlined inclusion/exclusion criteria above, 71 articles were ultimately included in the scoping review (Fig 1). The characteristics of the 71 included studies are summarized in Table 2.

**Table 2. Characteristics of the studies included in the scoping review grouped by setting (i.e., community-based/outpatient clinic or hospital).**

| Study | Country | Study Design | Patient Population[a] | Sample Size | Mean Age (± SD), years | Male, n (%) | ECG Data Collection Method |
|---|---|---|---|---|---|---|---|
| colspan | | | | | | | |
| *Community-Based and/or Outpatient Clinic (by year earliest to most recent)* | | | | | | | |
| Chiang 1969 [33] | US | Longitudinal epidemiological | Cardiac disease (+) | 264 | 30+ | NR | Single channel (duration unspecified) |
| Desai 1973 [34] | US | Observational | Cardiac disease (+/−) | 1,037 | Mean 60 | 640 (62%) | 12-lead for ≥30 seconds |
| Ruberman 1977 [35] | US | Observational | History of MI | 1,739 | Range 35–74 | 1,739 (100%) | Single-lead for 1-hour |
| Boudoulas 1979 [36] | US | Observational | Cardiac disease (+) | 339 | Mean 52 (19–72) | 223 (66%) | 24-hour Holter |
| Rabkin 1981 [37] | Canada | Observational | IHD (−) | 401 | Predominantly 25–35 | 401 (100%) | NR |
| Abdalla 1987 [38] | US | RCT | Apparently healthy | 15,481 | 46 ± 7 | 15,481 (100%) | 2-minute lead I rhythm strip |
| Dabrowski 1998 [39] | Poland | Observational | Cardiac symptoms (+) | 303 | 59 ± 14 | 215 (71%) | 12-lead |
| Dabrowski 1999 [40] | Poland | Observational | Post MI | 193 | 62 ± 10 | 161 (83%) | 12-lead |
| Jouven 2000 [41] | France | Prospective observational | CHD (−) | 6,101 | 48 ± 2 | 6,101 (100%) | Bipolar lead ($V_5$ and $V_5R$) per bicycle exercise test protocol |
| Hatanaka 2002 [42] | Japan | Observational | PVC (+), history of cardiac disease (−) | 201 | Mean 44 | 192 (96%) | 12-lead |
| Morshedi-Meibodi 2004 [43] | US | Observational | Cardiac disease (−) | 2,885 | 43 ± 10 | 1,385 (48%) | Leads $V_1$ and $V_5$ (during treadmill and recovery) |
| Carrim 2005 [44] | UK | Observational | ICD (+) | 44 | Mean 57 | 38 (86%) | ICD (duration unspecified) |
| Massing 2006 [45] | US | Observational | Cardiac disease (−) | 15,070 | 54 ± 0.1 | 6,766 (45%) | 2-minute leads V1, II, and $V_5$ rhythm strip; and 12-lead |
| Meine 2006 [46] | US | Observational | Referred for stress test; cardiac disease (−) | 2,828 | Median 58 (50–67) | 2,036 (72%) | 12-lead |
| Sajadieh 2006 [47] | Denmark | Epidemiological survey | Apparently healthy | 678 | 64 ± 7 | 398 (59%) | 48-hour two-channel Holter |
| Topaloglu 2007 [48] | Turkey | Observational | Cardiac symptoms (+) | 63 | 39 ± 9 | 19 (30%) | 24-hour Holter |
| Niwano 2009 [49] | Japan | Observational | Frequent PVCs; cardiac disease (−) | 239 | 43 ± 13 | 118 (49%) | Holter (duration unspecified) |
| Agarwal 2010 [50] | US | Observational | Cardiac disease (−) | 14,783 | 54 ± 6 | 6,652 (45%) | 2-minute leads $V_1$, II, and $V_5$ rhythm strip; and 12-lead |
| Baman 2010 [51] | US | Retrospective observational | Frequent PVC referred for ablation | 174 | 48 ± 13 | 87 (50%) | 24-hour Holter |
| Hirose 2010 [52] | Japan | Cohort | Cardiac disease (−) | 11,158 | 55 ± 11 | 4,333 (39%) | 12-lead |
| Le 2010 [53] | US | Observational | Cardiac disease (+) | 352 | 64 ± 11 | 345 (98%) | 12-lead |
| Cheriyath 2011 [54] | US | Observational | Cardiac disease (−) | 14,574 | 54 ± 6 | 6,327 (43%) | 2-minute leads $V_1$, II, and $V_5$ rhythm strip; and 12-lead |
| Munoz 2011 [55] | US | Observational | Frequent and symptomatic PVCs | 70 | 42 ± 17 | 30 (43%) | 24- to 48-hour Holter and 12-lead |
| Agarwal 2012 [56] | US | Observational | Cardiac disease (−) | 13,486 | 54 ± 6 | 5,932 (44%) | 2-minute leads $V_1$, II, and $V_5$ rhythm strip; and 12-lead |

*(Continued)*

**Table 2.** (Continued)

| Study | Country | Study Design | Setting | Patient Population | Sample Size | Mean Age (± SD), years | Male, n (%) | ECG Data Collection of PVCs |
|---|---|---|---|---|---|---|---|---|
| Ofoma 2012 [57] | US | Observational | Cardiac disease (–) | | 14,493 | Mean 54 | 6,232 (43%) | 2-minute leads $V_1$, II, and $V_5$ rhythm strip; and 12-lead |
| Yokokawa 2012a [58] | US | Observational | Frequent PVCs referred for ablation | | 241 | 48 ± 14 | 115 (48%) | 24-hour Holter |
| Yokokawa 2012b [59] | US | Observational | Frequent PVCs referred for ablation | | 294 | 48 ± 14 | 137 (47%) | 12-lead ECG (mapping procedure) and 24-hour Holter |
| Ban 2013 [60] | South Korea | Observational | Frequent PVCs underwent ablation | | 127 | 44 ± 13 | 50 (39%) | 24-hour Holter and 12-lead |
| Ephrem 2013 [61] | US | Cohort | Cardiac symptoms (+) | | 222 | 55 ± 16 | 94 (43%) | 24-hour three-channel Holter (unspecified) |
| Barutcu 2014 [62] | Turkey | Observational | | Frequent PVCs; SHD (–) | 80 | Mean 47 (22–60) | 34 (43%) | 24-hour Holter |
| Cozma 2014 [63] | Romania | Cross-sectional | | Cardiac disease (–) | 121 | 43 ± 11.5 | 76 (63%) | 24-hour Holter |
| Lee 2014 [64] | US | Observational | Cardiac disease (+/–) | | 1,589 | 61 ± 16 | 879 (55%) | 24-hour Holter and 12-lead |
| Park 2014 [65] | South Korea | Observational | Frequent PVCs; SHD (–) | | 438 | 55 ± 15 | 144 (33%) | 24-hour Holter |
| Pol 2014 [66] | US | Observational | Cardiac disease (+) | | 45 | 49 ± 15 | 17 (38%) | 24-hour Holter and 12-lead |
| Qureshi 2014 [67] | US | Cohort | Cardiac disease (–) | | 7,504 | 59 ± 13 | 3,977 (53%) | 12-lead |
| Yang 2014 [68] | US | Cross-sectional | High-burden PVC | | 66 | 64 ± 16 | 38 (58%) | 24-hour Holter and 12-lead |
| Agarwal 2015 [69] | US | Observational | Baseline stroke (–) | | 24,460 | 64 ± 9 | 10,990 (45%) | 12-lead |
| Dukes 2015 [70] | US | Cohort | Cardiac disease (–) | | 1,139 | Median 70 (68–74) | 482 (42%) | 24-hour Holter |
| Lin 2015 [71] | Taiwan | Observational | Apparently normal hearts with symptoms | | 3,351 | 58 ± 19 | 1,910 (57%) | 24-hour Holter |
| Bas 2016 [72] | US | Observational | Cardiac diseases (+/–) | | 107 | 50 ± 15 | 58 (54%) | 24-hour Holter |
| Nguyen 2017 [73] | US | Cohort | Cardiac disease (–) | | 18,910 | Range 49–80 | NR | 12-lead |
| Agarwal 2017 [74] | US | Longitudinal | Cardiac disease (–) | | 16,807,903 | 50 ± 19 | 7,091,590 (42%) | NR |
| Lin 2017 [75] | Taiwan | Observational | Apparently normal hearts with symptoms | | 5,778 | 63 ± 18 | 3,352 (61%) | 24-hour three-channel Holter (unspecified) |
| Sheldon 2017 [76] | US | Observational | Frequent idiopathic PVCs referred for ablation | | 100 | 52 ± 15 | 53 (53%) | 24-hour Holter |
| Aviles-Rosales 2018 [77] | Mexico | Observational | Cardiac disease (+) | | 1,442 | 56 ± 14 | 1,179 (82%) | Continuous 12-lead (treadmill and recovery) |
| Bière 2018 [78] | France | Observational | Probability of CAD | | 343 | 67 ± 12 | 235 (68.5%) | Continuous 12-lead (treadmill and recovery) |
| Li 2018 [79] | China | Observational | ICD (+) | | 416 | Median 64 (55–73) | 314 (76%) | ICD for 60-days |
| Ruwald 2018 [80] | US, Canada, Europe (multicenter) | RCT | CRT-D (+) | | 698 | 64 ± 10 | 520 (74%) | 24-hour Holter |
| Su 2018 [81] | China | Observational | Cardiac diseases (+/–) | | 342 | 57 ± 15 | 153 (45%) | 24-hour Holter |
| Altıntaş 2019 [82] | Turkey | Cross-sectional | Idiopathic PVC | | 341 | Median 50 (38–60) | 172 (50.4) | 24-hour Holter and 12-lead |
| **Study** | **Country** | **Study Design** | **Setting** | **Patient Population** | **Sample Size** | **Mean Age (± SD), years** | **Male, n (%)** | **ECG Data Collection of PVCs** |

(*Continued*)

**Table 2.** (Continued)

| | | | | | | | | |
|---|---|---|---|---|---|---|---|---|
| *In-Hospital (by year earliest to most recent)* | | | | | | | | |
| Schulze 1977 [83] | US | Observational | Non-ICU | AMI | 81 | Mean 56 | 61 (75%) | 24-hour Holter |
| Moss 1979 [84] | US | Observational | CCU | AMI | 940 | 54 ± 8 | 770 (82%) | 6-hour Holter |
| Rengo 1979 [85] | Italy | Observational | Non-ICU | Cardiac disease (+/−) | 232 | Mean 64 | 155 (67%) | 12-lead |
| Sclarovsky 1979 [86] | Israel | Observational | CCU | AMI | 114 | NR | NR | 12-lead |
| Lichtenberg 1980 [87] | US | Observational | Cath | AMI | 81 | NR | NR | Leads I, aVF, $V_1$ oscilloscopic recorder |
| Dash 1983 [88] | US | Observational | Cath | Cardiac disease (+) | 58 | NR | NR | 2-minute multiple lead (unspecified) |
| Bigger 1984 [89] | US | Observational | CCU | Post MI | 819 | Under 70 | NR | 24-hour leads $V_1$ and $V_5$ Holter |
| Kostis 1987 [90] | US | RCT | CCU | AMI | 1,640 | NR | NR | 24-hour two-channel Holter (unspecified) |
| Minisi 1988 [91] | US | Observational | Non-ICU | Post MI | 128 | 55 ± 10 | 91 (71%) | 24-hour Holter |
| Rasmussen 1988 [92] | Denmark | Observational | CCU | Post MI | 51 | Median 66 (24–89) | 33 (65%) | 24-hour two-channel Holter (unspecified) |
| Moulton 1990 [93] | US | Observational | Non-ICU | Cardiac disease (+) | 100 | 60 ± 2 | NR | 12-lead |
| Wilson 1992 [94] | US | RCT | CCU | AMI | 2,456 | NR | NR | 24-hour two-channel Holter (unspecified) |
| Fujimoto 1994 [95] | Japan | Observational | Non-ICU | Cardiac disease (+/−) | 100 | 58 ± 16 | 59 (59%) | Holter (unspecified) |
| Statters 1996 [96] | UK | Observational | CCU | AMI | 680 | 58 ± 9 | 537 (79%) | 24-hour leads II and modified CM5 Holter |
| Vaage-Nilsen 1998 [97] | Denmark | RCT | Non-ICU | Post MI | 250 | 60 ± 9 | 187 (75%) | 12- to 24-hour Holter |
| Dabrowski 2000 [98] | Poland | Observational | Non-ICU | Post MI | 148 | 61 ± 9 | 127 (86%) | 12-lead |
| Im 2018 [99] | South Korea | Observational | Non-ICU | Cardiac disease (−) | 373 | 60 ± 16 | 168 (45%) | 24-hour Holter |
| Yamada 2019 [100] | Japan | Observational | Non-ICU | Cardiac disease (+) | 435 | 65 ± 14 | 271 (62%) | 24-hour Holter |
| *Other Settings (by year earliest to most recent)* | | | | | | | | |
| Fries 2003 [101] | Germany | Observational | NR | ICD (+) | 38 | 58 ± 14 | 33 (87%) | ICD (unspecified) |
| Engel 2007 [102] | US | Observational | Both In- and Outpatient | AF or paced rhythm (−) | 45,401 | 56 ± 15 | 40,861 (90%) | 12-lead |
| Kanei 2008 [103] | US | Observational | NR | Frequent PVCs; SHD (−) | 108 | 50 ± 16 | 34 (31%) | 24-hour Holter |

12-lead = standard 12-lead ECG; AF = atrial fibrillation; AMI = acute myocardial infarction; CAD = coronary artery disease; Cath = cardiac catheterization lab; CCU = coronary care unit; CHD = coronary heart disease; CRT-D = cardiac resynchronization therapy with defibrillator; ECG = electrocardiogram; ICD = implantable cardioverter defibrillator; ICU = intensive care unit; IHD = ischemic heart disease; MI = myocardial infarction; NR = not reported; PVC = premature ventricular complex; RCT = randomize control trial; SD = standard deviation; SHD = structural heart disease; UK = United Kingdom; US = United States.

[a] (+) and (-) indicates whether the study populations do or do not have the listed criteria. For example, "cardiac disease (+)" means the study participants have cardiac disease, and "cardiac disease (+/-)" means the study includes participants with and without cardiac disease.

## Study characteristics

**Location, design and setting.** Of the 71 studies included, 39 (55%) were conducted in the United States (US), 16 (23%) from Asia (Japan, South Korea, Turkey, China, Taiwan, and Israel), 13 (18%) from Europe (Denmark, France, Germany, Italy, Poland, Romania, and United Kingdom), one (3%) from each Canada and Mexico. One study was a multi-national study including participants from the US, Canada, and Europe. Nearly all (n = 66, 93%) were observational; and five (7%) were secondary analyses from randomized clinical trials including the: Multiple Risk Factor Intervention Trial (MRFIT) [38]; Beta Blocker Heart Attack Trial (BHAT) [90, 94]; Danish Verapamil Infarction Trial II (DAVIT II) [97]; and Multicenter Automatic Defibrillator Implantation Trial with Cardiac Resynchronization Therapy (MADIT-CRT) [80]. Fifty (70%) of the studies were conducted in outpatient and/or ambulatory clinics. Several were secondary data analyses from large community-based observational and/or epidemiological studies including the Tecumseh Community Health Study [33], the Framingham Offspring Study [43], Healthcare Cost and Utilization Project (HCUP) [74], the Atherosclerosis Risk in Communities (ARIC) Study [45, 50, 54, 56, 57, 73], the Cardiovascular Health Study (CHS) [70, 73], the Reasons for Geographic and Racial Differences in Stroke (REGARDS) Study [69], and the Third National Health and Nutrition Examination Survey (NHANES III) [67]. A small number (n = 18, 25%) were conducted in the hospital setting at varied locations including: cardiac catheterization and electrophysiology laboratory; coronary care unit; and non-ICU units. Three (4%) of the studies did not fall within the community/outpatient or hospital setting, two did not report study setting and one included both in- and outpatient settings.

**Sampling criteria.** The study sample sizes ranged from <100 participants [44, 48, 55, 62, 66, 68, 83, 87, 88, 92, 101] to >16-million [74]. In the majority of the studies, data were obtained from medical records, while others collected data during clinic visits, hospital stay, and/or a combination of both locations. Several of the studies examined the significance of PVCs in a cohort of healthy adults who did not have a history of a cardiac disease(s). For example, the ARIC Study excluded participants not in sinus rhythm and those with cardiac rhythm disorders such as Wolff-Parkinson-White (WPW) syndrome, atrial fibrillation (AF)/atrial flutter, wandering atrial pacemaker, and supraventricular tachycardia (SVT) [45, 50, 54, 56, 57, 73]. In addition to rhythm disorders, several other secondary analyses using the ARIC Study database also excluded participants with heart failure (HF) [56], coronary heart disease (CHD) [54, 56, 57], or a history of stroke [50, 54, 57]. One study using the HCUP dataset excluded adults with both systolic and/or diastolic HF at enrollment, arrhythmogenic right ventricular dysplasia, paroxysmal ventricular tachycardia, and valvular heart disease [74]. Similarly, one study using the NHANES III dataset also excluded participants with known cardiovascular disease, ECG evidence of prior MI, paced rhythms, or AF [67].

In contrast, several studies purposefully examined patients with known cardiac disease. Initial studies done in the 1970's and 1980's were focused on hospitalized patients with acute MI [35, 83, 84, 86, 87, 89–92]. However, beginning in the late 2000's, the focus shifted to patients with frequent PVCs who were referred for catheter ablation [58–60, 72, 76], or patients with exercise induced-PVCs who later had a cardiac catheterization procedure [46]. Other studies investigated the significance of PVCs among patients with dual-chamber ICDs [44, 79, 80, 101], or patients with palpitations, syncope, near-syncope who were referred for Holter recording evaluation due to these clinical symptoms [61, 71, 75].

The mean age of the samples included was primarily between 50 and 60 years old. In 34 (48%) of the 71 studies, men made up the majority of the sample and four studies (6%) included only male participants [35, 37, 38, 41]. A small number of the studies did not report

demographics such as age (n = 5; 7%) or sex (n = 9; 13%). As mentioned previously, very few studies reported ethnicity or race. S3 Table shows details of the sampling criteria.

**ECG data collection method used to identify PVCs.** The ECG method used to identify PVCs varied considerably across the studies, and in two studies the ECG method was not reported [37, 74]. Of the 71 studies, 30 (42%) used a Holter recorder, with varied recording times (6 to 48-hours); 15 studies (21%) used a standard 12-lead ECG; while 10 studies (14%) used a short duration ECG rhythm strip (2-minutes to 1-hour). In a handful of studies, other methods were used including; both a standard 12-lead and Holter recording, ICD device, and ECG data obtained during a diagnostic procedure such as cardiac mapping or stress test.

## Key findings

**Diagnostic value of PVCs acute myocardial infarction.** Three studies examined the diagnostic importance of PVCs in the early phase of acute MI for identifying the location, or diagnosis of acute MI, heart region (ventricle) and QRS morphology of the PVCs [86–88]. For example, one study showed that a right bundle branch block (RBBB) pattern PVC(s) during the first 48 hours post-MI along with right axis deviation was correlated with anterior MI (anteroseptal and/or anterolateral), whereas a RBBB pattern PVC(s) and left axis deviation was associated with infero-posterior wall MI [86]. The QRS characteristics of the PVC, such as the duration of the Q-wave and/or Q/R ratio were found to have low sensitivity, high specificity and moderate positive predictive value (PPV) for identifying anterior, or inferior MI location [87, 88]. PVCs with a qR or qRS configuration also had low sensitivity but high specificity and PPV in the diagnosis of MI [88]. Details of the study results are provided in Table 3.

**Prognostic value of PVCs.** The vast majority of the studies examined the prognostic value of PVCs using a longitudinal design. The studies examined a multitude of clinical outcomes (described below) from different settings (i.e., outpatient, community-based, and hospital based) among patients with and without known cardiac disease (Table 4). The prognostic outcomes examined included: (1) SHD; (2) lethal arrhythmias; (3) AF and/or stroke; (4) IHD and other adverse outcomes; (5) all-cause mortality; and (6) cardiovascular mortality. A summary of each outcome category is described in the text below.

**Table 3. Diagnostic value of premature ventricular complexes (PVCs) during the early phase of acute myocardial infarction.**

| Study | Diagnostic Value | PVC Criteria | Notes |
|---|---|---|---|
| Sclarovsky 1979 [86] | Location of MI | • RBBB PVC pattern in $V_1$ and RAD is indicative of anterior wall AMI.<br><br>• RBBB PVC pattern in $V_1$ and LAD is indicative of infero-posterior wall AMI. | PVCs were considered acute when they appeared only during the first 48-hour of the acute phase of the infarction. Using a standard 12-lead ECG, the origin of the PVCs was determined primarily based on right- versus left-bundle branch block PVC pattern in $V_1$ and axis deviation criteria in leads II, III, and aVF |
| Lichtenberg 1980 [87] | Location of MI | • Lead $V_1$: a Q wave of ≥0.04 sec and a Q/R ratio >0.1 may detect anterior MI (sensitivity 0.36, specificity 0.87, and PPV 0.67).<br><br>• Lead aVF: a Q wave of ≥0.02 sec may detect abnormal inferior wall motion (sensitivity 0.22, specificity 0.94, and PPV 0.70). | PVCs induced by catheter stimulation in lead $V_1$ for the diagnosis of anterior wall MI, and lead aVF for the inferior wall MI |
| Dash 1983 [88] | Diagnosis of MI | Morphologic analysis of PVC has 29% sensitivity but high specificity (97%) and PPV (86%) for the diagnosis of MI. | Pre-defined PVC criteria for MI diagnosis: a qR or qRS configuration with a q-wave duration of at least 0.04 second |

AMI = acute myocardial infarction; ECG = electrocardiogram; LAD = left axis deviation; MI = myocardial infarction; PPV = positive predictive value; PVC = premature ventricular complex; RAD = right axis deviation; RBBB = right bundle branch block.

**Table 4. Prognostic significance of premature ventricular complexes (PVCs) based on the setting, patient population, and PVC criteria.**

| Prognostic Outcome | Patient Population [a] | PVC Criteria | Follow-Up (year) [b] |
|---|---|---|---|
| **Structural heart disease** | | | |
| *Community/Outpatient* | | | |
| Decreased LVEF | No cardiac disease | Presence of any | 10 [73] |
| HF | | | 5–15 [56, 73, 74] |
| LA dysfunction | | Frequent/high burden | NR [63, 65] |
| LV dysfunction | | | 1.2 [60] |
| Decreased LVEF | | | 5–13 [49, 70, 82] |
| CMP | | | 2.5–6.3 [51, 58, 59] |
| HF | | | 13.7 [68, 70] |
| HF | With cardiac disease/symptoms | Presence of any | 10 [71, 75] |
| LV dysfunction | | Frequent/high burden | NR [48] |
| CMP | | | 1.2 [66] |
| Decreased LVEF | | | NR [55] |
| *In-hospital* | | | |
| Decreased LVEF | With cardiac disease/symptoms | Presence of any | NR [93] |
| **Lethal arrhythmia** | | | |
| *Community/Outpatient* | | | |
| VT/VF/SCD | With cardiac disease/symptoms | Presence of any | 2.2 [39] |
| VT/VF | | Frequent/high burden | 2.2–3.6 [44, 79, 80] |
| VT/VF | | During exercise test | NR [36] |
| Life-threatening arrhythmia combined outcome | | | 14 [77] |
| *In-hospital* | | | |
| VT/VF/SCD | With cardiac disease/symptoms | Presence of any | 2.9 [98] |
| **Atrial fibrillation or stroke** | | | |
| *Community/Outpatient* | | | |
| Atrial fibrillation | No cardiac disease | Presence of any | 10–15 [50, 73] |
| Stroke | | | 6–15 [50, 57, 69] |
| Atrial fibrillation and TIA | With cardiac disease/symptoms | Presence of any | 10 [71] |
| *In-hospital* | | | |
| Stroke-like symptoms | No cardiac disease | High burden | 3.5 [99] |
| **Coronary heart disease or other adverse cardiac outcomes** | | | |
| *Community/Outpatient* | | | |
| Ablation outcome (≥80% reduction of baseline PVC burden) | No cardiac disease | Presence of any | 5.6 mo [76] |
| Develop IHD | | | 10–14 [37, 54] |
| Myocardial ischemia | | During exercise testing | 4.6 [46] |
| Hospitalization | With cardiac disease/symptoms | Presence of any | 10 [71, 75] |
| Combined adverse events (ACS, stroke, CHF, all-cause mortality) | | | 2.3 [61] |
| *In-hospital* | | | |
| CAD severity | With cardiac disease/symptoms | Frequent | NR [91] |
| Cardiac event (ICD therapy, re-hospitalization) | | | 2.3 [100] |
| **Mortality** | | | |
| *Community/Outpatient* | | | |
| All-cause mortality | No cardiac disease | Presence of any | 10–13 [45, 67] |
| | | Frequent/high burden | 4.4–7 [38, 47] |
| | | During exercise testing | 15 [43] |

*(Continued)*

**Table 4.** (Continued)

| Prognostic Outcome | Patient Population [a] | PVC Criteria | Follow-Up (year) [b] |
|---|---|---|---|
| All-cause mortality | With cardiac disease/symptoms | Presence of any | 2–10 [33, 35, 40, 71, 75] |
| | | Frequent/high burden | 2.2–3.6 [79, 80] |
| | | During exercise test | 4.5–14 [53, 77, 78] |
| *In-hospital* | | | |
| All-cause mortality | With cardiac disease/symptoms | Presence of any | 7 mo– 2.3 [83, 90, 92, 94] |
| | | Frequent | 1–4.7 [89, 96, 97] |
| **Cardiovascular mortality** | | | |
| *Community/Outpatient* | | | |
| CV mortality | No cardiac disease | Presence of any | 11.9 [52] |
| Sudden cardiac death | | | 7–14 [38, 54] |
| Sudden cardiac death | | Frequent/high burden | 7 [38] |
| CV mortality | | During exercise testing | 15–23 [41, 43] |
| Sudden cardiac death | With cardiac disease/symptoms | Presence of any | 2 [35] |
| Cardiac death | | Frequent/high burden | 3.6 [79] |
| CV mortality | | During exercise test | 6.2 [53] |
| *In-hospital* | | | |
| Cardiac death | With cardiac disease/symptoms | Presence of any | 3 [84] |

Studies with no details on the setting, or studies that included participants with *AND* without cardiac disease (combined) are not included in the table. ACS = acute coronary syndrome; CAD = coronary artery disease; CV = cardiovascular; CHF = congestive heart failure; CMP = cardiomyopathy; HF = heart failure; ICD = implantable cardioverter defibrillator; IHD = ischemic heart disease; LA = left atrial; LV = left ventricular; LVEF = left ventricular ejection fraction; NR = not reported; PVC = premature ventricular complex; SCD = sudden cardiac death; TIA = transient ischemic attack; VF = ventricular fibrillation; VT = ventricular tachycardia.

[a] **Patient Population Categories**: No cardiac disease includes apparently healthy individuals, without ischemic/structural heart disease, idiopathic PVCs; cardiac symptoms include syncope, lightheadedness/near-syncope, dizziness, palpitations, angina, and/or dyspnea.

[b] Represents the follow-up period range using mean/median years (or months [mo] when indicated). References not reporting the follow-up timeframe [36, 44, 48, 55, 62, 63, 65, 82, 91, 93]. Details on follow-up period for all studies are available in S3 Table.

*Structural heart disease.* Patients with frequent PVCs have a larger left atrium (LA) diameter and volume index compared to those without frequent PVCs [62, 65]. LA remodeling suggests higher filling pressures in the LV, which impacts the LA shape and volume due to increased filling pressure and can indicate the presence of diastolic dysfunction and/or HF [63]. For example, one study showed that among patients with an extreme number of PVCs (>1,000/24-hour) along with symptoms (palpitations or dyspnea), the odds of having a trapezoidal LA was 1.32 higher for each 10% increase in PVC frequency (odds ratio, OR, 1.32, 95% confidence interval, CI, 1.17–1.48) [63]. This study also found that a high frequency of PVCs was associated with a higher LA volume [63].

The presence of PVCs has also been associated with LV dysfunction. In patients followed for 5-years, the presence of PVCs measured from a baseline ECG was associated with 2.8-times greater odds of having a reduced LV ejection fraction (LVEF) as compared to patients without PVCs [70, 73]. PVC morphology has also been associated with a lower LVEF. For example, patients with a lower LVEF have PVCs that are either notched or have a wide shelved QRS [93], or have a greater coupling interval and longer PVC QRS duration [64, 82]. Studies show that there is a higher prevalence of LV dysfunction in patients with frequent PVCs (≥ 10 PVCs/hour or > 1,000 PVCs/day) [49, 103] or a high burden of PVCs (i.e., PVCs >30% of all beats/day) [60], indicating PVCs are a marker of LV dysfunction. One study reported that for each 1% incremental increase in the daily rate of PVCs, there was an

increased risk of impaired LV relaxation (diastolic dysfunction) (OR 1.18, 95% CI 1.02–1.37) [48]. Another study found that PVCs that originate from the right ventricle (RV) were associated with a lower LVEF when the PVC burden was at least 10%, while PVCs originating from the left ventricle (LV) were associated with a reduced LVEF when the PVC burden was at least 20% [55]. Finally, among patients with frequent PVCs who are asymptomatic, a 10% increase in PVC burden was found to be an independent predictor of impaired LV function (OR 2.1, 95% CI 1.2–3.6) [58].

A number of studies found that PVCs were associated with the development of cardiomyopathy (CMP), so called PVC-induced CMP (PVC-CMP), and/or HF [56, 68, 70, 71, 73–75]. In patients referred for catheter ablation due to frequent idiopathic PVCs, those with a high PVC burden (>24% over 24-hour) were at increased risk for developing CMP [51, 58, 59, 72]. The QRS duration (width) of the PVCs had been found to be an independent predictor of PVC-CMP even when controlling for other important variables, such as symptoms, PVC origin, and PVC burden [59, 66, 72]. Several studies showed that wider QRS's increased the risk for PVC-CMP from 3% to 12% [51, 59, 72]. Data from large cohort studies (ARIC [56, 73], CHS [70, 73], and HCUP [74]) showed that there is an increased risk of incident HF (1.3- to 2-fold) when PVCs are present on either a 2-minute ECG rhythm strip or standard 12-lead ECG. In a sub-group analysis of individuals < 65 years of age without hypertension, diabetes mellitus (DM), CHD, or AF, there was a higher risk for incident HF compared to participants without PVCs (hazard ratio, HR, 6.5, 95% CI 5.5–7.7) [74]. In patients referred for 24-hour Holter monitoring due to syncope, palpitations, or suspected arrhythmia, there was a 1.5-fold increase in the rate new-onset HF when multiform PVCs were present compared to no PVCs (HR 1.46, 95% CI 1.06–2.00) [71]. Finally, patients with a high PVC burden ($\geq$ 20%) were found to have 3 times the odds of developing HF compared to the control group (OR 3.15, 95% CI 1.28–6.50) [68].

*Lethal arrhythmia(s).* Several studies have examined the relationship between PVCs and the development of VT and/or VF [36, 39, 44, 77, 79–81, 98, 101]. In patients with frequent PVCs (> 6/minute), high PVC burden ($\geq$ 10% of QRS's/24-hour) and/or couplets, were associated with an increased risk for VT (PVC burden OR = 1.07, 95% CI 1.03–1.11; PVC couplets OR = 33.98, 95% CI 11.53–100.19) [81]. In post-MI patients and those admitted for rule-out cardiac diagnosis, longer QT dispersion of the PVCs, defined as the difference between maximum and minimum QT interval of the PVCs measured across the 12-leads, was associated with the occurrence of lethal arrhythmias [39, 98]. In a study in patients with exercise-induced PVCs (EiPVCs) followed for up to 14 years, there was an increased risk of VF/flutter and/or sustained VT [77]. Another study showed that PVCs that occur in the late-period of exercise testing and/or frequent multiform PVCs was associated with VT or VF [36]. In patients with reduced LV function or CMP, a high PVC burden (>10 /hour) increased the risk for VT/VF by 2.8-fold (HR 2.79, 95% CI 1.69–4.58) [80]. In patients with an ICD, the frequency of PVCs was higher among those who developed VT/VF compared to those who did not develop VT/VF [44, 79]. One study examined R-on-T type PVCs in a group of patients with an ICD and found that this type of PVC rarely induced sustained VT. However, in the small number of patients with sustained VT induced by an R-on-T, the PVCs were more likely to be polymorphic (positive and negative QRS) and in patients with CAD [101].

*Atrial fibrillation and stroke.* In two large cohort studies, there was a 1.1- to 1.6-fold increased risk of developing AF when PVCs were present on an ECG recording [50, 73], although the study by Nguyen et al. also examined premature atrial complex (PAC) [73]. In one study that included patients with syncope, palpitations, or suspected arrhythmia, but who did not have a history of documented heart disease, the presence of multiform PVCs was associated with 1.5-times higher risk of new-onset AF compared to those without PVCs (HR 1.5,

95% CI 1.06–2.26) [71]. In two different community-based studies, researchers found an association between the presence of any PVCs (vs. absence of PVCs) and an increased risk for incident stroke at 6-year (HR 1.4, 95% CI 1.05–1.81) [69] and 15-year follow-up periods (HR 2.1, 95% CI 1.2–3.6) [50]. A subsequent analysis in one of these studies showed that the occurrence of $\geq$ 4 PVCs/minute was associated with a higher risk of stroke (HR 2.06, 95% CI 1.24–3.42) when compared to patients who did not have PVCs [50]. Additionally, when comparing any PVCs vs. no PVCs, non-hypertensive individuals were at increased risk for thrombotic stroke of non-carotid origin (HR 3.48, 95% CI 1.74–6.95). Interestingly, this same association was not present among patients with hypertension (HR 1.21, 95% CI 0.58–2.53) [50]. Similarly, in a secondary analysis of a longitudinal study (13 years) using data from the ARIC study (n = 14,493), participants with PVCs who were normotensive had a higher risk of ischemic stroke compared to hypertensive individuals with PVCs (HR 1.69, 95% CI 1.02–2.79) [57].

Among a group of patients without history of cardiac disease referred for 24-hour Holter recording for new cardiac symptoms (i.e., syncope, palpitations, suspected arrhythmia, and/or other clinical indication), those with multiform PVCs had an increased risk for transient ischemic attack at 10-years follow-up (HR 1.41, 95% CI 1.06–1.87) [71]. In a 3.5-years of follow-up period, frequent PVCs, defined as >10% of the total number of beats over 24-hours, were associated with the occurrence of stroke-like symptoms (i.e., painless weakness, sudden numbness, "dead" feeling on one side of the body, sudden but painless loss of vision, and sudden loss of the ability to understand what people were saying) (OR 3.42, 95% CI 1.09–10.73) [99].

*Coronary heart disease and other adverse cardiac outcomes.* Community based studies show that the presence of PVCs is correlated with a 1.2-fold increased risk for the development of ischemic/CAD over a 10-year follow-up period [37, 54]. Moreover, frequent PVCs ($\geq$0.6 PVCs/hour) following MI were an independent predictor of CAD severity [91]. Similarly, PVCs during the recovery phase of a stress test also predicted myocardial ischemia on myocardial perfusion imaging, adjusting for stress test scores and other significant clinical predictors (OR 1.27, 95% CI 1.04–1.56) [46].

In a 10-year follow-up study in patients with palpitations and syncope with suspected arrhythmia, the presence of >12 PVCs/day was found to increase a person's risk for cardiovascular-related hospitalization (crude HR 1.24, 95% CI 1.06–1.45) [75]. In two separate studies, multiform PVCs (vs. no PVCs) were associated with all-cause hospitalization (HR 1.20, 95% CI 1.06–1.35) [71], cardiovascular-related hospitalization (HR 1.29, 95% CI 1.03–1.61) [71], and a major adverse cardiovascular event or new/worsening HF (HR 3.05, 95% CI 1.39–6.70) [61]. In a study that followed hospitalized patients with decompensated HF for up to 2 years, PVC burden (per 1% increase) was an independent risk factor for a cardiac event, such as ICD therapy, re-hospitalization, or death (HR 1.036; CI 1.005-1.068) [100]. In a different analysis, patients with left bundle branch block (LBBB) type PVCs (negative QRS in lead $V_1$) without axis deviation had a lower incidence of various cardiac diseases (i.e., hypertension, IHD, CMP, and/or valvular heart disease) than those with RBBB-type PVCs [42]. Finally, in a study of patients with frequent idiopathic PVCs who were referred for catheter ablation, the presence of pleomorphic PVCs (i.e., multiple morphologies in at least three ECG leads) with a cut-off of $\geq$156 PVCs/24-hour was associated with a non-successful ablation outcome [76].

*All-cause mortality.* In population-based studies with follow-up from 3.5 and 13 years, the presence of PVCs at enrollment was associated with an increased risk of all-cause mortality [33, 34, 37, 45]. In another community-based study, $\geq$30 PVCs/hour during a 48-hour Holter recording was associated with all-cause mortality, or a first acute MI event (HR 2.46, 95% CI 1.29–4.68) [47]. Among 1,139 participants from the CHS Study, individuals with frequent PVCs (upper versus lowest quartile) had a 1.31-fold increased risk of death during a median follow-up of 13 years [70]. When considering the potential mediation effect of HF, the

presence of PVCs on a 12-lead ECG was associated with an increased risk of overall mortality during the 10-year follow-up period among ARIC Study participants (HR 1.5, 95% CI 1.3–1.8), but this was not significant in CHS Study participants (HR 1.1, 95% CI 0.9–1.20) [73]. Interestingly, in another large study (n = 7,504), the presence of PVCs at enrollment was an independent risk factor of mortality only among individuals ≥65 years (HR 1.36, CI 1.04–1.76) [67].

Rengo et al. showed that in 232 hospitalized patients followed for 30-months who had at least one PVC on a standard 12-lead ECG, those with a short coupling interval of <360 milli-second had a higher rate of sudden death [85]. Similarly, hospitalized patients who had PVCs on any resting ECG had a doubled of mortality (HR 2.0, 95% CI 1.1–2.8) over a 5.5-year follow-up period [102]. In other studies, PVCs during exercise stress testing (EiPVCs) were associated with mortality in patients without cardiovascular and/or valvular heart disease [43], and in patients with known CAD or idiopathic CMP [77] during the 14-year follow-up period. Moreover, among patients referred for single-photon emission computed tomography (SPECT), the presence of EiPVCs was a risk factor for mortality in patients with preserved (> 50%) LVEF (OR 2.17, 95% CI 1.09–4.34) [78]. Multiform PVCs and >12 PVCs/day during a 24-hour Holter recording increased the risk of mortality by 1.5-fold among patients with suspected arrythmia with syncope and palpitations during 10-year follow-up [71, 75]. Similarly, in patients with cardiac resynchronization therapy and a defibrillator (CRT-D), high PVC burden (>10 PVCs/hour) was associated with increased risk of HF and death (HR 2.76, p<0.001) [80].

Numerous studies have examined the association between PVCs and mortality in patients with MI [35, 40, 83, 89, 90, 92, 94, 96, 97]. In a study with only male MI participants, the presence of complex PVCs (i.e., "early" PVC or R-on-T, run-PVCs, multiform PVC, or bigeminy) vs. no complex PVCs was associated with a higher risk of mortality (RR 1.9, p ≤ 0.01) [35]. In patients with PVCs on a 12-lead ECG recorded at least 4 weeks post-MI, a PVC QT dispersion ≥100 millisecond was an independent predictor of mortality (HR 3.10, 95% CI 1.7–9.4) [40]. In patients admitted to a coronary care unit for acute MI, several studies showed that PVCs were associated with an increased risk of death at both 1- and 2-year follow-up [89, 90, 92, 94, 96]. Two of these studies showed that the risk of mortality was nearly 2-fold in patients with a run of PVCs (≥2 consecutive) [89] and 3.6-fold in patients with complex PVCs (i.e., couplets, multiform, run of PVCs, and R-on-T) [92]. Those treated with thrombolytics who had ≥ 25 PVCs/hour had a higher risk of mortality as compared with those who had <25 PVCs/hour [96]. Similarly, those treated without thrombolytics who had ≥ 10 PVCs/hour had a higher risk of mortality as compared with those who had< 10 PVCs/hour [96].

*Cardiovascular mortality.* In apparently healthy adult populations, the presence of PVCs at enrollment to the study was associated with 2- to 3.7-fold increased risk for cardiac death measured up to at least a 4-year [38, 47, 52, 54]. In particular, frequent PVCs defined as ≥30/hour (vs. <30 PVCs/hour) significantly increased the risk for cardiovascular mortality (HR 2.85, 95% CI 1.16–7.0) [47]. Similarly, frequent PVCs during exercise stress testing was associated with a higher risk of cardiovascular death over a 15-year follow-up time frame [41, 43]. In patients with clinical HF, the presence of PVCs increased the risk of cardiac death by 5.48-fold [53]; while high PVC burden (i.e., ≥40% in 60-days continuous monitoring) was associated with a higher risk of cardiac mortality in patients with a dual-chamber ICD (HR 3.29, 95% CI 1.72–6.28) [79]. In the setting of MI, patients with PVCs had a 2.8-fold increased risk of sudden cardiac death as compared with those who did not have PVCs [35].

## Discussion

To our knowledge, this is the first scoping review to carefully map the available evidence on the diagnostic and prognostic significance of PVCs across different care settings (community versus hospital) and patient populations (with and without heart disease). Our review, covering half a century (1969 to 2019), included predominantly observational studies mostly in the outpatient and/or community-based settings. These studies show that PVCs and associated characteristics (e.g., frequency, burden, and morphology) are associated with adverse outcomes, although the spectrum of risk is quite varied.

Three hospital-based studies from the early 1980s examined the diagnostic significance of PVCs in patients with acute MI. These studies were designed to better understand the occurrence of PVCs in the early phase of acute MI when ECG monitoring was relatively new in hospital settings. Two of the studies compared whether PVCs were associated with the diagnosis of MI, comparing PVC morphology from a standard 12-lead ECG to findings during cardiac catheterization, and found poor sensitivity but higher specificity and PPV for acute MI [87, 88]. One of these studies showed that patients with a qR or qRS PVC morphology and a q-wave duration of at least 0.04 seconds on their ECG have a high probability of having an acute MI (86% PPV) [88]. These studies, however, are outdated and even if replicated do not appear relevant given the many advances in MI diagnostic methods (i.e., biomarkers, cardiac imaging) and treatments used in current practice. One line of inquiry that might be useful is an examination of the clinical significance of PVCs during continuous ECG monitoring given that these data are readily available from bedside or telemetry ECG monitors, which have become ubiquitous in the hospital setting. These studies should include both cardiac and non-cardiac patients and examined in both ICU and non-ICU settings given that this has not been studied. Such a study could help guide whether careful monitoring of PVCs during continuous ECG monitoring using alarms (audible or inaudible) is important for identifying high risk patients.

Early studies focused primarily on the association between PVCs, measured during hospitalization, and all-cause and cardiovascular mortality measured after discharge, particularly among patients with acute MI. The goal was to investigate whether treating or suppressing PVCs would prevent future adverse patient outcomes. The Cardiac Arrhythmia Pilot Study (CAPS), a double-blind RCT in 502 acute MI patients enrolled between 6- and 60-days post-MI with at least 10 PVCs/hour, was the first study to test whether suppression of PVCs with antiarrhythmic drugs would improve survival [104]. At 12-months, the investigators showed that encainide, flecainide, and moricizine, suppressed ventricular arrhythmias and were well tolerated by the study participants [104]. Based on these findings, these drugs were tested in the much larger CAST study to test whether suppressing PVCs in patients with asymptomatic or mildly symptomatic ventricular arrhythmias would reduce mortality [15]. Surprisingly, preliminary results showed a higher rate of death from an arrhythmia and/or non-fatal cardiac arrest (relative risk, RR, 3.6, 95% CI 1.7–8.5), as well as total mortality (RR 2.5, 95% CI 1.6–4.5) in the encainide and flecainide group compare to the placebo group [15]. Based upon these findings, aggressive anti-arrhythmic drug treatment was no longer recommended [15, 105]. Following the CAST study, researcher shifted their focus from the post-MI population to the general population. Although initially PVCs were believed to be a marker of disease severity, recent population-based studies showed that PVCs appear to have a significant role in the development of adverse outcomes when measured longitudinally. Among healthy adults, the presence of PVCs increased the risk for AF, stroke, LA and LV dysfunction, IHD, and all-cause and cardiac mortality. Similar findings were also observed in patients with cardiac disease or symptoms suggestive of cardiac disease. Moreover, in these patients there is an

associated risk of PVCs and arrhythmia events (i.e., sustained VT, VF, or sudden cardiac death) and hospitalization.

One particular finding of interest from this review is that the presence of PVCs is linked with AF [50, 71, 73] and ischemic stroke [50, 57, 69]. This is physiologically plausible given that PVCs can occur in the presence of SHD, particularly LV pathology. Atrial stretch or remodeling due to LV dysfunction can lead to AF, which can cause the blood clots associated with stroke. However, AF has also been associated with premature atrial beats arising from the pulmonary veins, thus, PVCs alone are not likely to explain this association. However, only one study [73] also examined atrial ectopy when assessing the correlation between PVCs and AF. In addition, other studies [50, 57, 69] did not account for the potential mediation of antiarrhythmic medications when examining the association of PVCs and incident stroke, potentially introducing bias. It is worth noting that all of these studies were community based. To our knowledge, there are no study that have examined whether PVCs during hospitalization might be correlated with new-onset AF or stroke post-discharge. This could be a rational for monitoring for PVCs during hospitalization to identify patients at risk for future events.

Another finding of interest in this scoping review were studies that examined the association of PVCs and the occurrences of lethal arrhythmias (i.e., VT/VF). Of note, the studies included in our review examined this correlation years after discharge and not in the hospital setting. For example, in one study that included hospitalized MI patients, lethal arrhythmia events were measured at three years [98]. In a community-based study in patients with an ICD, arrhythmia events were examined at a median of 3.5 years [44, 79]. PVC monitoring during hospital-based ECG monitoring was initiated decades ago for identifying patients at risk for lethal arrhythmias and remains a central rationale today. Surprisingly, we did not find any published hospital-based studies examining the prognostic significance of PVCs and lethal arrhythmias. To date, only case reports or studies with small sample sizes have been published [106–110]. Although the mechanisms of how PVCs trigger lethal arrhythmias in some patients is not entirely understood, PVCs have been associated with VF [25], thus, treatment of idiopathic VF using PVC ablation techniques has been found to be effective [111–113]. Further research is needed to examine not only associations but potential cause and effect relationships of PVCs and lethal arrhythmias in both community- and hospital-based populations.

Recently, there is a growing interest in the significance of PVCs and the development of CMP and incident HF. Researchers began examining the potential role of PVCs in the development of CMP after early reports showed that improved LVEF was seen after suppressions of PVCs in patients with dilated CMP [114–116]. As shown in the studies in this review, frequent PVCs or high PVC burden appear to be a risk factor in the development of PVC-CMP. In practice, radiofrequency ablation has been shown to improve LV function especially in patients who have high PVC burden [25, 117], and is now recommended as a first line treatment for PVCs [23, 118]. Unfortunately, there is a lack of prospective studies that could explain the exact mechanisms that underlying PVC-CMP. Similarly, to date, there is a lack of clinical trials to show whether PVCs cause HF, and almost all of the available evidence on these topics comes from cross-sectional and longitudinal studies. Cross-sectional studies suggested that the PVCs might be a manifestation for the extent of HF [68, 82]. Longitudinal studies might provide a better insight into the association between PVCs and LV dysfunction since PVCs are present before incident HF or LV dysfunction [56, 70, 73, 74]. There could be factors other than PVCs that confound an individual's likelihood for developing HF or LV dysfunction that might not be captured in cross-sectional studies. Interestingly, interventional studies show improvement in LV function after radiofrequency catheter ablation treatment of PVCs, particularly in patients with frequent PVCs [119–122]. Although it appears that PVCs are associated with LV dysfunction, these studies were not designed to determine the causal effect of

PVCs on LV dysfunction, thus future studies are needed. Nonetheless, these studies show a link between PVCs, LV dysfunction and HF.

Mortality has been one of the major outcomes of interest examined across studies in relation to PVCs since the 1980s. Our review shows that PVCs are associated with mortality when measured at long-term follow-up among patients with and without cardiac disease, similar to previously published meta-analyses [13, 26, 27]. The association between PVCs and mortality, however, requires careful interpretation, especially when considering the follow-up period across the available studies. As shown in Table 4, in the community/outpatient setting, the association between PVCs and mortality was measured after at least 2-years. In hospital-based studies, a similar association was found after one-year follow-up. Although these studies suggest that PVCs are a risk factor for mortality, the aforementioned studies are observational, thus, a causal relationship has not been established. Currently, there is no convincing data available to show that treatment of PVCs reduces mortality.

Other areas of research related to this topic would benefit from further investigation. For example, EiPVCs remain a topic of debate and, unfortunately there remains uncertainty as to the significance of these types of PVCs in clinical practice. A recent meta-analysis showed an association between EiPVCs in asymptomatic patients, particularly PVCs during the recovery period, and adverse cardiovascular events (e.g., angina, non-fatal MI, cardiac hospital admission, or cardiac arrest) at long-term follow-up (5.5 to 16 years) [123]. The authors suggest that autonomic dysfunction may play a role in this association. However, the analysis included only observational studies. Therefore, additional studies are needed to determine if interventions (i.e., drug or catheter-based) can improve the prognosis of patients with EiPVCs.

Another topic of interest is the clinical relevance of PVCs in hospitalized patient populations, which surprisingly has not been studied in any meaningful way. This topic has important implications for continuous ECG monitoring because PVC alarms are common [7–9]. One observational study reported that PVCs were the most prevalent arrhythmia type alarm during continuous bedside ECG monitoring [7]. Of the over 2.5 million unique alarms in 461 ICU patients during a one-month period, there were 854,901 (33%) PVC alarms. However, to our knowledge there are no published contemporary studies that have examined the association of PVCs and arrhythmias or adverse patient outcomes in the hospital setting. When ECG monitoring was introduced in cardiac ICUs in the 1960s, PVC were carefully monitored for and treated with antiarrhythmic drugs because they were believed to be associated with lethal arrhythmias [124, 125]. Following the CAST Study published in 1989 showing that treatment was actually harmful [15], aggressive treatment of PVCs was no longer standard practice. Yet, PVC alarms are still activated (i.e., turned on) in ECG monitors used in hospital units with ECG devices (i.e., ICU, step-down, medical surgical). Given the high number of PVC alarms generated during continuous ECG monitoring, it is safe to assume that alarm burden/fatigue is exacerbated among nurses and providers from this type of alarm. Moreover, there is scant evidence to support the clinical benefit of continuously monitoring for PVCs in the acute/critical care setting. Of note, many manufacturers offer PVC algorithms for many types of PVCs (e.g., isolated, bigeminy, trigeminy, couplets, R-on-T, or VT>2); thus, adding to the complexity of monitoring for PVCs and alarm burden. Future studies are needed to examine the clinical significance of these PVCs in the hospitalized patients and whether enabling PVC alarms from continuous ECG monitoring informs clinical care.

Finally, it is important to note that almost all of the studies in this scoping review were observational; therefore, a causal relationship between PVCs and adverse patient outcomes remains largely unknown. Comprehensive identification of PVCs using the ECG, while non-invasive, can be somewhat challenging from both a technical and patient burden perspective and are likely reflected in the study designs included in this review. For example, in a number

of the studies, a snap shot 12-lead ECG was used to identify PVCs. Other ECG methods used were 24- to 48-hour Holter recordings and in three studies PVCs were identified with an ICD. All of these methods, with the exception of the ICD studies, represent relatively short time windows of PVC assessment. Given newer ECG technology and wearables sensors that can be worn for long periods of time with minimal patient burden could open up opportunities for not only more comprehensive identification of PVCs, but as a way to assess interventions. Although the studies included in this review provide important information on the clinical significance of PVCs, there is still much to be learn about translating this knowledge into clinical practice. For example, there are limited guideline-based data on the best approach for health-care providers to take when encountering a patient with a PVC(s) on their 12-lead ECG or PVCs during continuous ECG monitoring. With the growing evidence showing the potential predictive value of PVCs and certain outcomes (e.g., CMP and HF), there is also a need for studies to validate the predictive value of PVCs and its utilization to mitigate the potential adverse consequences of PVCs.

## Limitations

We acknowledge that this review has limitations. First, as mentioned above, a number of the studies included in this scoping review are dated, with some from the 1970s to 1990s. There-fore, the outcomes of these studies, which were influenced by clinical management at the time, might not reflect current patient populations and/or management of PVCs in current clinical practice. Nevertheless, this scoping review highlights the significant paradigm shifts over decades concerning the clinical significance of PVCs. In the 1970s, PVCs were thought to have a significant role in the development of lethal arrhythmias and risk of mortality, and therefore, were aggressively treated with antiarrhythmic drugs. This practice changed following the pub-lication of the CAST study in the early 1990s, thus, shifting the paradigm away from aggres-sively treating PVCs. By the turn of the 21$^{st}$ century, however, researchers showed a plausible association between PVCs and long-term poor outcomes, which has led to new PVC treatment strategies in select patients. Given that patients are living longer and have co-occurring cardiac co-morbidities, the impact of PVCs remains a relevant research topic in both community and in-patient settings.

Second, we limited our search to studies published in English from four bibliographic data-bases. Therefore, it is possible we missed studies that have been published in other languages, or reports that are published outside of traditional peer-reviewed commercial publications (grey literature). To account for grey literature, we searched the reference lists of available arti-cles included.

Third, very few studies reported demographic and/or socioeconomic data. In addition, the majority of the studies included mostly male participants, which could limit our knowledge on whether sex differences play an important role in the association of PVCs and poor patient outcomes. How PVCs might impact varied populations remains largely unknown and should be explored in broader populations. For example, according to 2021 Heart Disease and Stroke Statistics from the American Heart Association, the CVD mortality rate was highest among non-Hispanic Black or African American people compared to other racial and ethnic groups [126]. In this same report, Black or African American individuals 20 to 54 years of age had a higher annual sex-adjusted incidence of first-ever ischemic stroke (128 per 100,000) than White individuals (48 per 100,000). Similar disparities, while not as dramatic as seen among Blacks or African Americans, are also observed among Native Americans, Hispanics and Asian/Pacific Islanders when compared to their White counterparts. Based on a 2012 study, between 2010 and 2050, the number of incident strokes is expected to more than double, with

the majority of the increase among the elderly ($\geq$75 years of age) and minority groups. Given the findings of the studies included in this scoping review showing the association of PVCs to cardiovascular diseases, additional research is needed that examines PVCs in the aforementioned populations.

Finally, we were unable to perform a critical appraisal or examine publication bias because of significant between-study design heterogeneity, research setting, patient population(s), ECG data collection method, analysis approach and adverse outcomes of interest. Furthermore, almost all of the included studies were observational, and therefore, clear causal relationships between PVCs and patient outcomes remains unclear. However, because this was a scoping review, heterogeneity was expected as our goal was to describe the available evidence on this topic without paying strict attention to homogeneity across studies as one would do in a systematic review. Despite this, our approach allowed us to examine this topic in a much broader way, across settings (community- and hospital-based) and among patient populations with- and without cardiac diseases, which will guide future research on this topic.

## Conclusion

A very small number of dated studies, all hospital-based, have examined the diagnostic value of PVCs without validation in contemporary patient populations; hence, no clear associations can be drawn. However, there have been a number of published studies, in large cohorts followed for years, on the prognostic value of PVCs. These studies were conducted across different care settings (community and hospital), and in patients with and without cardiovascular diseases. The findings from these studies show that PVCs are not entirely benign. PVCs morphologies (e.g., multiform PVC, LBBB- or RBBB-type), frequency/burden, and setting observed in, are associated with long-term adverse cardiovascular outcomes. This scoping review identified areas for future research including: improved PVC assessment techniques (i.e., newer ECG devices and sensor technology); evaluation in people of color, women, and older individuals; hospital-based PVC monitoring; and studies designed to examine whether there is a cause and effect association of PVCs with adverse patient outcomes.

## Supporting information

**S1 Table. Preferred reporting items for systematic reviews and meta-analysis extension for scoping review (PRISMA-ScR) checklist.**
(DOCX)

**S2 Table. Database search results.**
(DOCX)

**S3 Table. Methodology and key findings of the included studies.**
(DOCX)

## Author Contributions

**Conceptualization:** Sukardi Suba.

**Data curation:** Sukardi Suba.

**Formal analysis:** Sukardi Suba.

**Investigation:** Michele M. Pelter.

**Methodology:** Sukardi Suba, Kirsten E. Fleischmann, Hildy Schell-Chaple, Priya Prasad, Gregory M. Marcus, Xiao Hu, Michele M. Pelter.

**Resources:** Sukardi Suba, Kirsten E. Fleischmann, Hildy Schell-Chaple, Priya Prasad, Gregory M. Marcus, Xiao Hu, Michele M. Pelter.

**Supervision:** Kirsten E. Fleischmann, Hildy Schell-Chaple, Priya Prasad, Gregory M. Marcus, Xiao Hu, Michele M. Pelter.

**Validation:** Sukardi Suba, Michele M. Pelter.

**Visualization:** Sukardi Suba.

**Writing – original draft:** Sukardi Suba.

**Writing – review & editing:** Sukardi Suba, Kirsten E. Fleischmann, Hildy Schell-Chaple, Priya Prasad, Gregory M. Marcus, Xiao Hu, Michele M. Pelter.

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
