## [Decision Letter · Decision Letter 0]

6 Oct 2021

PONE-D-21-25033Diagnostic and prognostic significance of premature ventricular complexes in community and hospital-based participants: A scoping reviewPLOS ONE

Dear Dr. Suba,

Thank you for submitting your manuscript to PLOS ONE. After careful consideration, we feel that it has merit but does not fully meet PLOS ONE’s publication criteria as it currently stands. Therefore, we invite you to submit a revised version of the manuscript that addresses the points raised during the review process.

Especially, please highlight in the limitations the historical context of the cohort. 

We look forward to receiving your revised manuscript.

Kind regards,

Antonio Cannatà

Academic Editor

PLOS ONE

Journal Requirements:

5. We note that this manuscript is a systematic review or meta-analysis; our author guidelines therefore require that you use PRISMA guidance to help improve reporting quality of this type of study. Please upload copies of the completed PRISMA checklist as Supporting Information with a file name “PRISMA checklist”.

Reviewers' comments:

Reviewer's Responses to Questions

**Comments to the Author**

1. Is the manuscript technically sound, and do the data support the conclusions?

Reviewer #1: Partly

Reviewer #2: Partly

Reviewer #3: Yes

2. Has the statistical analysis been performed appropriately and rigorously? 

Reviewer #1: No

Reviewer #2: N/A

Reviewer #3: N/A

3. Have the authors made all data underlying the findings in their manuscript fully available?

Reviewer #1: Yes

Reviewer #2: Yes

Reviewer #3: Yes

4. Is the manuscript presented in an intelligible fashion and written in standard English?

Reviewer #1: Yes

Reviewer #2: Yes

Reviewer #3: Yes

5. Review Comments to the Author

Reviewer #1: Suba and colleagues produced a scope review, analyzing many articles available in the literature and at the end evaluating the diagnostic and prognostic significance of premature ventricular complexes (PVCs) in 71 articles, mostly made up of observational studies.

The study is well written and deals with a very interesting and debated topic. However, the work has some concerning aspects:

-The authors analyzed very dated works (even from 1970s and 1980s). This, in my opinion makes the job weaker from a prognostic point of view; many heart diseases, especially the ischemic cardiopathy, had a huge improvement in prognosis. This topic should be analyzed focusing on more recent studies.

-The topic of PCVs is extremely heterogeneous and the prognosis in patients with frequent PVCs is affected by the presence of underlying cardiac disease. Treating the topic as a whole, albeit differentiated by categories (structural heart disease, ischemic heart disease, etc.), reduces its significance. It maybe would have been appropriate to analyze only one topic or analyzing the individual subgroups more precisely (type of heart disease, setting from which the patients came, described in more detail).

- Being PVCs a very frequent ECG anomaly, it is important to consider only high quality and meaningful works, to fully understand the subject and to formulate precise concepts that are prognostically meaningful.

-To facilitate the reading, it would have been useful to have a table with summaries of the main concepts divided by topic

-Exercise-induced PVCs are a matter of debate. A paragraph on the subject would have been interesting.

-In the paragraph "structural heart disease" the authors could deepen the topic, analyzing subgroups of diseases that cause left ventricular dysfunction, such as cardiomyopathies, which are diseases with arrhythmic burden that often cause management controversies.

Reviewer #2: The authors describe a study where they performed a scoping review of published studies to determine the diagnostic and prognostic significance of PVCs in the community and hospital settings and amongst patients with and without cardiac disease. Despite significant heterogeneity between studies, the authors report a number of findings from their review including the presence of PVCs associated with a larger LA volume, LV dysfunction, lethal arrhythmias, increased risk of all-cause mortality and cardiovascular mortality. There is very little new information to be gained from this work - with the possible exception of the observation that PVCs were associated with an increased risk of stroke and AF. This type of study does not determine causality between these associations. Although the breadth of literature reviewed is vast, the authors note that most studies reviewed were observational and there is limited value in the conclusions that the authors have drawn - over and above what is already known.

Reviewer #3: This is an interesting scoping review about diagnostic and prognostic significance of PVCs across different care settings and patient populations covering half a century (1969 to 2019). Although the signficant between-study heterogeneity in designs, setting, patient populations, ECG data collection method, analysis approach authours succeed describe the available evidence on this topic. The review appeares complete and clear. It is well written and discussion is appropriate. The scoping review highlights need for for future research examining the clinical significance of and best practices regarding PVCs especially in acute setting. I would like to congratulate the authors on their work.

6. PLOS authors have the option to publish the peer review history of their article (what does this mean?). If published, this will include your full peer review and any attached files.

Reviewer #1: No

Reviewer #2: **Yes: **Rahul Mukherjee

Reviewer #3: No

---

## [Author Response · Author response to Decision Letter 0]

18 Nov 2021

Dear Editor and Reviewers, 

Thank you for reviewing our manuscript entitled “Diagnostic and prognostic significance of premature ventricular complexes in community and hospital-based participants: A scoping review” and providing thoughtful comments, questions, and suggestions. We have carefully reviewed the comments and suggestions and have revised the manuscript accordingly. Per instructions, revisions have been highlighted on the re-submitted manuscript. In addition, below are our responses (in red) to each of the reviewers' comments. 

Academic Editor: 

Journal Requirements:

Response:

We apologize for the formatting errors in our initial submission and thank you for the opportunity to re-format the paper. We believe the paper now conforms to the journal’s requirements.

Response:

As noted above we have re-formatted the submission by adding tables as part of the main manuscript and have removed individual table files from the submission platform. Supplementary tables remain uploaded) as separate "supporting information" files.

Response:

Thank you for pointing this out. We will update this information during the re-submission process.

Response:

Thank you for pointing this out. The corresponding author for the editorial process has been changed from Sukardi Suba (first author) to Michele Pelter (last author) to ensure that the corresponding author is affiliated with the chosen institution. 

5. We note that this manuscript is a systematic review or meta-analysis; our author guidelines therefore require that you use PRISMA guidance to help improve reporting quality of this type of study. Please upload copies of the completed PRISMA checklist as Supporting Information with a file name “PRISMA checklist”.

Response:

Please note a PRISMA checklist, in our case PRISMA for a scoping review (PRISMA-ScR), was submitted as a supplement with the original version (see S1 Table).

Response: 

 Captions for the Supporting Information has been added to the end of the manuscript. 

Thank you for submitting your manuscript to PLOS ONE. After careful consideration, we feel that it has merit but does not fully meet PLOS ONE’s publication criteria as it currently stands. Therefore, we invite you to submit a revised version of the manuscript that addresses the points raised during the review process.

Especially, please highlight in the limitations the historical context of the cohort. 

Response: 

Dr. Cannatà, thank you for the opportunity to revise our manuscript. In response to your comment regarding highlighting the historical context of the studies included in the limitations, we have added the following:

Page 36 – 37 lines 601 – 614 (also highlighted in the re-submission): 

We acknowledge that this review has limitations. First, as mentioned above, a number of the studies included in this scoping review are dated, with some from the 1970s to 1990s. Therefore, the outcomes of these studies, which were influenced by clinical management at the time, might not reflect current patient populations and/or management of PVCs in current clinical practice. Nevertheless, this scoping review highlights significant paradigm shifts over decades concerning the clinical significance of PVCs. In the 1970s, PVCs were thought to have a significant role in the development of lethal arrhythmias and risk of mortality, and therefore, were aggressively treated with antiarrhythmic drugs. This practice changed following the publication of the CAST study in the early 1990s, thus, shifting the paradigm of aggressively treating PVCs. By the turn of the 21st century, however, researchers showed a plausible association between PVCs and long-term outcomes, which has led to new PVC treatment strategies in select patients. Given that patients are living longer and have co-occurring cardiac co-morbidities, the impact of PVCs remains a relevant research topic in both community and in-patient settings. 

Reviewer #1: 

Suba and colleagues produced a scope review, analyzing many articles available in the literature and at the end evaluating the diagnostic and prognostic significance of premature ventricular complexes (PVCs) in 71 articles, mostly made up of observational studies.

The study is well written and deals with a very interesting and debated topic. However, the work has some concerning aspects:

-The authors analyzed very dated works (even from 1970s and 1980s). This, in my opinion makes the job weaker from a prognostic point of view; many heart diseases, especially the ischemic cardiopathy, had a huge improvement in prognosis. This topic should be analyzed focusing on more recent studies. 

Thank you for taking the time to carefully review our manuscript and providing comments and feedback to improve its quality. We acknowledge that this review included very dated works. A primary motivation in conducting this scoping review was based on data from our research team examining alarm fatigue in clinicians, particularly nurses, due to the excessive number of alarms during continuous ECG monitoring. In a one-month time period, PVC alarms accounted for 33% (854,000) of 2.5 million total alarms (n=461 ICU patients). We hypothesize that the vast majority of PVC alarms are not clinically important and contribute to alarm fatigue. However, before testing PVC alarm reduction interventions or making alarm adjustment recommendations, we wanted to better understand the clinical relevance of PVCs and the basis of PVC monitoring - a standard practice during ECG monitoring in the ICU setting. Therefore, we purposely did not limit the publication dates of articles included to capture a broader understanding of this topic and highlight the paradigm shift(s) from carefully monitoring for and treating PVCs to a more conservative and focused treatment approach as is practiced today. We included both out- and in-patient studies to examine whether out patient studies might help identify important patient factors that might justify hospital-based PVC monitoring. To better describe the rationale behind this we have edited the manuscript in the following ways:

1. Abstract has been re-written:

Lines 25 to 45:Background. While there are published studies that have examined premature ventricular complexes (PVCs) among patients with and without cardiac disease, there has not been a comprehensive review of the literature examining the diagnostic and prognostic significance of PVCs. This could help guide both community and hospital-based research and clinical practice. Methods. Scoping review frameworks by Arksey and O’Malley and the Joanna Briggs Institute (JBI) were used. A systematic search of the literature using four databases (CINAHL, Embase, PubMed, and Web of Science) was done. The review was prepared adhering to the Preferred Reporting Items for Systematic Reviews and Meta-Analysis Extension for Scoping Review (PRISMA-ScR). Results. A total of 71 relevant articles were identified, 66 (93%) were observational, and five (7%) were secondary analyses from randomized clinical trials. Three studies (4%) examined the diagnostic importance of PVC origin (left/right ventricle) and QRS morphology in the diagnosis of acute myocardial ischemia (MI). The majority of the studies examined prognostic outcomes including left ventricular dysfunction, heart failure, arrhythmias, ischemic heart diseases, and mortality by PVCs frequency, burden, and QRS morphology. Conclusions. Very few studies have evaluated the diagnostic significance of PVCs and all are decades old. No hospital setting only studies were identified. Community-based longitudinal studies, which make up most of the literature, show that PVCs are associated with structural and coronary heart disease, lethal arrhythmias, atrial fibrillation, stroke, all-cause and cardiac mortality. However, a causal association between PVCs and these outcomes cannot be established due to the purely observational study designs employed.

2. Introduction: Added to page 3 lines 52 – 54 the following: 

One hospital-based study found that there were over 854,901 PVC alarms during the one-month study period representing, 33% of the over 2.5 million total alarms or 18 PVCs/monitoring hour per patient.

3. End of Introduction page 5, lines 101 – 103: 

Of note, this scoping review included historical studies as far back as the 1970’s with the goal of understanding shifts in patient populations of study, clinical and prognostic importance and the analysis of newer cardiac pathologies associated with PVCs.

4. Discussion section (also highlighted in the resubmission):

a. Page 30, lines 462 – 470:

These studies, however, are outdated and even if replicated do not appear relevant given the many advances in MI diagnostic methods (i.e., biomarkers, cardiac imaging) used in current practice. One line of inquiry that might be useful, is an examination of the clinical significance of PVCs during continuous ECG monitoring given that these data are readily available from ECG monitors, which have become ubiquitous in the hospital setting. These studies should include both cardiac and non-cardiac patients and be examined in both ICU and non-ICU settings given that this has not been studied. Such a study could help guide whether careful monitoring of PVCs during continuous ECG monitoring using alarms (audible or inaudible) is important for identifying high risk patients.

b. Page 32, lines 502 – 506:

It is worth noting that all of these studies were community based. To our knowledge, there are no study that have examined whether PVCs during hospitalization might be correlated with new-onset AF or stroke post-discharge. This could be a rational for monitoring for PVCs during hospitalization to identify patients at risk for future events.

c. Page 32 – 33, lines 512 – 521:

PVC monitoring during hospital-based ECG monitoring was initiated decades ago for identifying patients at risk for lethal arrhythmias and remains a central rationale today. Surprisingly, we did not find any published hospital-based studies examining the prognostic significance of PVCs and lethal arrhythmias. To date, only case reports or studies with small sample sizes have been published. Although the mechanisms of how PVCs trigger lethal arrhythmias in some patients is not entirely understood, PVCs have been associated with VF, thus, treatment of idiopathic VF using PVC ablation techniques has been found to be effective. Further research is needed to examine not only associations but potential cause and effect relationships of PVCs and lethal arrhythmias in both community- and hospital-based populations.

d. Limitations section Page 36 – 37 lines 601 – 614:

First, as mentioned above, a number of the studies included in this scoping review are dated, with some from the 1970s to 1990s. Therefore, the outcomes of these studies, which were influenced by clinical management at the time, might not reflect current patient populations and/or management of PVCs in current clinical practice. Nevertheless, this scoping review highlights significant paradigm shifts over decades concerning the clinical significance of PVCs. In the 1970s, PVCs were thought to have a significant role in the development of lethal arrhythmias and risk of mortality, and therefore, were aggressively treated with antiarrhythmic drugs. This practice changed following the publication of the CAST study in the early 1990s, thus, shifting the paradigm of aggressively treating PVCs. By the turn of the 21st century, however, researchers showed a plausible association between PVCs and long-term outcomes, which has led to new PVC treatment strategies in select patients. Given that patients are living longer and have co-occurring cardiac co-morbidities, the impact of PVCs remains a relevant research topic in both community and in-patient settings. 

e. Limitation Section page 37 lines 620 – 634:

Third, very few studies reported demographic and/or socioeconomic data. In addition, majority of the studies included mostly male participants, which could limit our knowledge on whether sex differences played important role in the association of PVCs and patient outcomes. How PVCs might impact varied populations remains largely unknown and should be explored in broader populations. For example, according to 2021 Heart Disease and Stroke Statistics from the American Heart Association, the CVD mortality rate was highest among non-Hispanic Black people compared to other racial and ethnic groups. In this same report, Black individuals 20 to 54 years of age had a higher annual sex-adjusted incidence of first-ever ischemic stroke (128 per 100,000) than White individuals (48 per 100,000). Similar disparities, while not as dramatic as seen among Blacks, are also observed among Hispanics and Asian/Pacific Islanders when compared to their White counterparts. Based on a 2012 study, between 2010 and 2050, the number of incident strokes is expected to more than double, with the majority of the increase among the elderly (≥75 years of age) and minority groups. Given the findings of studies included in this scoping review showing the association of PVCs to these cardiovascular diseases additional research is needed that examines this question in the aforementioned populations.

f. Limitations Section page 38, lines 638 – 639:

Furthermore, almost all of the included studies were observational, and therefore, clear causal relationships between PVCs and patient outcomes remains unclear. 

5. Conclusions page 38 – 39, lines 646 – 657: 

A very small number of dated studies have examined the diagnostic value of PVCs (all in hospital-based setting) without validation from contemporary studies; hence, no clear associations can be drawn. However, there have been a number of studies, including those with large samples who were followed for years, published on the prognostic value of PVCs. These studies were conducted across different care settings (community and hospital), and in patients with and without cardiovascular diseases. The findings from these studies show that PVCs are not entirely benign. PVCs morphologies (e.g., multiform PVC, LBBB- or RBBB-type), frequency/burden, and setting observed in, are associated with long-term adverse cardiovascular outcomes. This scoping review identified areas for future research including: improved PVC assessment techniques (i.e., newer ECG devices and sensor technology); evaluation in people of color, women, and older individuals; hospital-based PVC monitoring; and studies designed to examine whether there is a cause and effect association of PVCs with adverse outcomes. 

-The topic of PCVs is extremely heterogeneous and the prognosis in patients with frequent PVCs is affected by the presence of underlying cardiac disease. Treating the topic as a whole, albeit differentiated by categories (structural heart disease, ischemic heart disease, etc.), reduces its significance. It maybe would have been appropriate to analyze only one topic or analyzing the individual subgroups more precisely (type of heart disease, setting from which the patients came, described in more detail).

Adding to the points made above, we designed our review to gain a broader understanding of the significance of PVCs. As seen in the results of our review, we identified indications for PVC monitoring that are less relevant in current practice (e.g., PVCs in diagnosis of myocardial infarction), those more clinically important (e.g., PVC induced cardiomyopathies, heart failure), and topics that require further investigation (e.g., exercise induced PVC, PVC-triggered ventricular tachycardia/fibrillation). 

- Being PVCs a very frequent ECG anomaly, it is important to consider only high quality and meaningful works, to fully understand the subject and to formulate precise concepts that are prognostically meaningful.

Thank you for pointing this out; we fully agree with this statement. However, the goal of this scoping review was to “map” available evidence on this topic, regardless of the quality. Per our scoping protocol, we intended to conduct a critical appraisal/publication bias assessment; however, after assessing the included studies, we determined it was not feasible to conduct a critical appraisal/bias assessment due to the methodological heterogeneity of the studies. We addressed this on page 8, lines 167-172.

-To facilitate the reading, it would have been useful to have a table with summaries of the main concepts divided by topic

Thank you for this excellent suggestion. We have revised Table 4, page 20, lines 272 to 287 using the headings by the prognostic outcomes examined in the paper (i.e., structural heart disease, lethal arrhythmia, atrial fibrillation or stroke, CHD or other adverse outcomes, mortality, and cardiovascular mortality). 

-Exercise-induced PVCs are a matter of debate. A paragraph on the subject would have been interesting.

Thank you for pointing this out. We have edited the paper in the following way:

Page 34, lines 553 – 560: 

Other areas of research related to this topic would benefit from further investigation. For example, EiPVCs remain a topic of debate and, unfortunately there remains uncertainty as to the significance of these types of PVCs in clinical practice. A recent meta-analysis showed an association between EiPVCs, particularly PVCs during the recovery period, and adverse cardiovascular events (e.g., angina, non-fatal MI, cardiac hospital admission, or cardiac arrest) at long-term follow-up (5.5 to 16 years). However, the analysis included ten studies, all were observational, thus, limiting generalizability of the findings. Therefore, additional studies are needed to advance our understanding regarding the mechanisms and clinical relevance of EiPVCs.

-In the paragraph "structural heart disease" the authors could deepen the topic, analyzing subgroups of diseases that cause left ventricular dysfunction, such as cardiomyopathies, which are diseases with arrhythmic burden that often cause management controversies.

Thank you. In the discussion section of the original paper (page 33, lines 530 – 541) we discuss LV dysfunction, cardiomyopathies, and heart failure. We have also added the following to address cardiomyopathies:

Discussion section page 33, lines 522 – 530:

Recently, there is a growing interest in the significance of PVCs and the development of CMP and incident HF. Researchers began examining the potential role of PVCs in the development of CMP after early reports that showed improved LVEF after suppressions of PVCs in patients with dilated CMP. As shown in the studies in this review, frequent PVCs or high PVC burden appear to be a risk factor in the development of PVC-CMP. In practice, radiofrequency ablation has been shown to improve LV function especially in patients who have high PVC burden, and is now recommended as a first line treatment for PVCs. Unfortunately, there is a lack of prospective studies that could explain the exact mechanisms that underlying PVC-CMP.

Reviewer #2: 

The authors describe a study where they performed a scoping review of published studies to determine the diagnostic and prognostic significance of PVCs in the community and hospital settings and amongst patients with and without cardiac disease. Despite significant heterogeneity between studies, the authors report a number of findings from their review including the presence of PVCs associated with a larger LA volume, LV dysfunction, lethal arrhythmias, increased risk of all-cause mortality and cardiovascular mortality. There is very little new information to be gained from this work - with the possible exception of the observation that PVCs were associated with an increased risk of stroke and AF. This type of study does not determine causality between these associations. Although the breadth of literature reviewed is vast, the authors note that most studies reviewed were observational and there is limited value in the conclusions that the authors have drawn - over and above what is already known.

Thank you for taking the time to carefully review our manuscript and providing feedback to improve the paper. A primary motivation in conducting this scoping review was based on data from our research team examining alarm fatigue in clinicians, particularly nurses, due to the excessive number of alarms during continuous ECG monitoring. In a one-month time period PVC alarms accounted for 33% (854,000) of 2.5 million total alarms (n=461 ICU patients). We hypothesize that the vast majority of PVC alarms are not clinically significant and contribute to alarm fatigue. However, before testing alarm reduction interventions to address PVC alarms, we wanted to better understand the clinical relevance of PVCs and the basis of PVC monitoring, which is standard practice during ECG monitoring in the ICU settings. We purposely did not limit the publication dates or study design of articles included to capture a broader understanding of this topic and highlight the paradigm shift from carefully monitoring for PVCs and treating them to a more conservative and focused treatment approach as is practiced today. We included both out- and in-patient studies to examine whether out patient studies might help identify important patient factors that might justify hospital-based PVC monitoring. To better describe the rationale behind this work we have edited the manuscript in the following ways:

1. Abstract has been re-written:

Lines 25 to 45:Background. While there are published studies that have examined premature ventricular complexes (PVCs) among patients with and without cardiac disease, there has not been a comprehensive review of the literature examining the diagnostic and prognostic significance of PVCs. This could help guide both community and hospital-based research and clinical practice. Methods. Scoping review frameworks by Arksey and O’Malley and the Joanna Briggs Institute (JBI) were used. A systematic search of the literature using four databases (CINAHL, Embase, PubMed, and Web of Science) was done. The review was prepared adhering to the Preferred Reporting Items for Systematic Reviews and Meta-Analysis Extension for Scoping Review (PRISMA-ScR). Results. A total of 71 relevant articles were identified, 66 (93%) were observational, and five (7%) were secondary analyses from randomized clinical trials. Three studies (4%) examined the diagnostic importance of PVC origin (left/right ventricle) and QRS morphology in the diagnosis of acute myocardial ischemia (MI). The majority of the studies examined prognostic outcomes including left ventricular dysfunction, heart failure, arrhythmias, ischemic heart diseases, and mortality by PVCs frequency, burden, and QRS morphology. Conclusions. Very few studies have evaluated the diagnostic significance of PVCs and all are decades old. No hospital setting only studies were identified. Community-based longitudinal studies, which make up most of the literature, show that PVCs are associated with structural and coronary heart disease, lethal arrhythmias, atrial fibrillation, stroke, all-cause and cardiac mortality. However, a causal association between PVCs and these outcomes cannot be established due to the purely observational study designs employed.

2. Introduction: Added to page 3 lines 52 – 54 the following: 

One hospital-based study found that there were over 854,901 PVC alarms during the one-month study period representing, 33% of the over 2.5 million total alarms or 18 PVCs/monitoring hour per patient.

3. End of Introduction page 5, lines 101 – 103: 

Of note, this scoping review included historical studies as far back as the 1970’s with the goal of understanding shifts in patient populations of study, clinical and prognostic importance and the analysis of newer cardiac pathologies associated with PVCs.

4. Discussion section (also highlighted in the resubmission):

a. Page 30, lines 462 – 470:

These studies, however, are outdated and even if replicated do not appear relevant given the many advances in MI diagnostic methods (i.e., biomarkers, cardiac imaging) used in current practice. One line of inquiry that might be useful, is an examination of the clinical significance of PVCs during continuous ECG monitoring given that these data are readily available from ECG monitors, which have become ubiquitous in the hospital setting. These studies should include both cardiac and non-cardiac patients and be examined in both ICU and non-ICU settings given that this has not been studied. Such a study could help guide whether careful monitoring of PVCs during continuous ECG monitoring using alarms (audible or inaudible) is important for identifying high risk patients.

b. Page 32, lines 502 – 506:

It is worth noting that all of these studies were community based. To our knowledge, there are no study that have examined whether PVCs during hospitalization might be correlated with new-onset AF or stroke post-discharge. This could be a rational for monitoring for PVCs during hospitalization to identify patients at risk for future events.

c. Page 32 – 33, lines 512 – 521:

PVC monitoring during hospital-based ECG monitoring was initiated decades ago for identifying patients at risk for lethal arrhythmias and remains a central rationale today. Surprisingly, we did not find any published hospital-based studies examining the prognostic significance of PVCs and lethal arrhythmias. To date, only case reports or studies with small sample sizes have been published. Although the mechanisms of how PVCs trigger lethal arrhythmias in some patients is not entirely understood, PVCs have been associated with VF, thus, treatment of idiopathic VF using PVC ablation techniques has been found to be effective. Further research is needed to examine not only associations but potential cause and effect relationships of PVCs and lethal arrhythmias in both community- and hospital-based populations.

d. Limitations section Page 36 – 37 lines 601 – 614:

First, as mentioned above, a number of the studies included in this scoping review are dated, with some from the 1970s to 1990s. Therefore, the outcomes of these studies, which were influenced by clinical management at the time, might not reflect current patient populations and/or management of PVCs in current clinical practice. Nevertheless, this scoping review highlights significant paradigm shifts over decades concerning the clinical significance of PVCs. In the 1970s, PVCs were thought to have a significant role in the development of lethal arrhythmias and risk of mortality, and therefore, were aggressively treated with antiarrhythmic drugs. This practice changed following the publication of the CAST study in the early 1990s, thus, shifting the paradigm of aggressively treating PVCs. By the turn of the 21st century, however, researchers showed a plausible association between PVCs and long-term outcomes, which has led to new PVC treatment strategies in select patients. Given that patients are living longer and have co-occurring cardiac co-morbidities, the impact of PVCs remains a relevant research topic in both community and in-patient settings. 

e. Limitation Section page 37 lines 620 – 634:

Third, very few studies reported demographic and/or socioeconomic data. In addition, majority of the studies included mostly male participants, which could limit our knowledge on whether sex differences played important role in the association of PVCs and patient outcomes. How PVCs might impact varied populations remains largely unknown and should be explored in broader populations. For example, according to 2021 Heart Disease and Stroke Statistics from the American Heart Association, the CVD mortality rate was highest among non-Hispanic Black people compared to other racial and ethnic groups. In this same report, Black individuals 20 to 54 years of age had a higher annual sex-adjusted incidence of first-ever ischemic stroke (128 per 100,000) than White individuals (48 per 100,000). Similar disparities, while not as dramatic as seen among Blacks, are also observed among Hispanics and Asian/Pacific Islanders when compared to their White counterparts. Based on a 2012 study, between 2010 and 2050, the number of incident strokes is expected to more than double, with the majority of the increase among the elderly (≥75 years of age) and minority groups. Given the findings of studies included in this scoping review showing the association of PVCs to these cardiovascular diseases additional research is needed that examines this question in the aforementioned populations.

f. Limitations Section page 38, lines 638 – 639:

Furthermore, almost all of the included studies were observational, and therefore, clear causal relationships between PVCs and patient outcomes remains unclear. 

5. Conclusions page 38 – 39, lines 646 – 657: 

A very small number of dated studies have examined the diagnostic value of PVCs (all in hospital-based setting) without validation from contemporary studies; hence, no clear associations can be drawn. However, there have been a number of studies, including those with large samples who were followed for years, published on the prognostic value of PVCs. These studies were conducted across different care settings (community and hospital), and in patients with and without cardiovascular diseases. The findings from these studies show that PVCs are not entirely benign. PVCs morphologies (e.g., multiform PVC, LBBB- or RBBB-type), frequency/burden, and setting observed in, are associated with long-term adverse cardiovascular outcomes. This scoping review identified areas for future research including: improved PVC assessment techniques (i.e., newer ECG devices and sensor technology); evaluation in people of color, women, and older individuals; hospital-based PVC monitoring; and studies designed to examine whether there is a cause and effect association of PVCs with adverse outcomes.

Reviewer #3: 

This is an interesting scoping review about diagnostic and prognostic significance of PVCs across different care settings and patient populations covering half a century (1969 to 2019). Although the signficant between-study heterogeneity in designs, setting, patient populations, ECG data collection method, analysis approach authours succeed describe the available evidence on this topic. The review appeares complete and clear. It is well written and discussion is appropriate. The scoping review highlights need for for future research examining the clinical significance of and best practices regarding PVCs especially in acute setting. I would like to congratulate the authors on their work.

Thank you for your comments. We hope that this review will provide insights for clinicians and researchers to advance our knowledge and understanding of this topic and areas that require further investigation.

---

## [Editor Report · Decision Letter 1]

9 Dec 2021

Diagnostic and prognostic significance of premature ventricular complexes in community and hospital-based participants: A scoping review

PONE-D-21-25033R1

Dear Dr. Pelter,

We’re pleased to inform you that your manuscript has been judged scientifically suitable for publication and will be formally accepted for publication once it meets all outstanding technical requirements.

Kind regards,

Antonio Cannatà

Academic Editor

PLOS ONE

---

## [Editor Report · Acceptance letter]

15 Dec 2021

PONE-D-21-25033R1 

Diagnostic and prognostic significance of premature ventricular complexes in community and hospital-based participants: A scoping review 

Dear Dr. Pelter:

I'm pleased to inform you that your manuscript has been deemed suitable for publication in PLOS ONE. Congratulations! Your manuscript is now with our production department. 

Kind regards, 

on behalf of

Dr. Antonio Cannatà 

Academic Editor

PLOS ONE